# Pan-Genome Plasticity and Virulence Factors: A Natural Treasure Trove for *Acinetobacter baumannii*

**DOI:** 10.3390/antibiotics13030257

**Published:** 2024-03-14

**Authors:** Theodoros Karampatakis, Katerina Tsergouli, Payam Behzadi

**Affiliations:** 1Microbiology Department, Papanikolaou General Hospital, 57010 Thessaloniki, Greece; tkarampatakis@yahoo.com; 2Microbiology Department, Agios Pavlos General Hospital, 55134 Thessaloniki, Greece; ktsergouli@gmail.com; 3Department of Microbiology, Shahr-e-Qods Branch, Islamic Azad University, Tehran 37541-374, Iran

**Keywords:** *Acinetobacter baumannii*, genome, plasticity, virulence factors, molecular pathogenicity

## Abstract

*Acinetobacter baumannii* is a Gram-negative pathogen responsible for a variety of community- and hospital-acquired infections. It is recognized as a life-threatening pathogen among hospitalized individuals and, in particular, immunocompromised patients in many countries. *A. baumannii,* as a member of the ESKAPE group, encompasses high genomic plasticity and simultaneously is predisposed to receive and exchange the mobile genetic elements (MGEs) through horizontal genetic transfer (HGT). Indeed, *A. baumannii* is a treasure trove that contains a high number of virulence factors. In accordance with these unique pathogenic characteristics of *A. baumannii*, the authors aim to discuss the natural treasure trove of pan-genome and virulence factors pertaining to this bacterial monster and try to highlight the reasons why this bacterium is a great concern in the global public health system.

## 1. Introduction

*Acinetobacter baumannii* is named after the American bacteriologists Linda and Paul Baumann. The most isolated strains of *Acinetobacter* spp. from clinical samples belong to *A. baumannii* [1]. According to the proposal of Brisou and Prévot, which was based on the results of transformation assays, the non-motile bacteria of *Achromobacter* were reclassified as *Acinetobacter* genus in 1954 [1,2].

In 1911, some surveys analyzed *Acinetobacter* isolates taken from the soil, which was known as *Micrococcus calcoaceticus*. As a new bacterial genus, *Acinetobacter* was officially classified in 1971 [3]. In accordance with the List of Prokaryotic names with Standing in Nomenclature (LPSN) (https://lpsn.dsmz.de/, accessed on 11 March 2024), the name *Acinetobacter* is rooted in the Greek language, meaning *A* (non-); *cinet* (motile); *bacter* (rod). With the progression of advanced high-throughput molecular technologies, 82 validly published species in association with the bacterium of *Acinetobacter* have been reported up to 11 March 2024 (https://lpsn.dsmz.de/genus/acinetobacter, accessed on 11 March 2024) [4].

Although the ubiquitous Gram-negative, catalase-positive, aerobic, non-fermenter coccobacilli of *A. baumannii* is recognized as a non-motile bacterium, it can be moved via a twitching process [3,5,6]. As a ubiquitous bacterium, *A. baumannii* can be recognized in different natural habitats and environments, e.g., soil and water. Due to this knowledge, this bacterium can also be isolated from non-human hosts such as arthropods (human head lice), animals and plants. In this regard, *A. baumannii* is known as an important pathogen among animals and in veterinary clinics [7,8].

*A. baumannii* is a life-threatening pathogen among hospitalized individuals and, in particular, immunocompromised patients in many countries. *A. baumannii* may cause serious hospital (HAIs) and community-acquired infections (CAIs), e.g., bacteremia, septicemia, severe (ventilator-associated) pneumonia, etc. 

Moreover, *A. baumannii* is a member of the ESKAPE group. ESKAPE is an acronym for *Enterococcus faecium*, *Staphylococcus aureus*, *Klebsiella pneumoniae*, *A. baumannii*, *Pseudomonas aeruginosa* and *Enterobacter* spp. [5,6,9,10]. As previous studies show, the ESKAPE group members are predisposed to receive and exchange the mobile genetic elements (MGEs), e.g., plasmids bearing antimicrobial resistance genes (ARGs) through horizontal genetic transfer (HGT). On the other hand, antimicrobial resistance (AMR) is a significant global concern and is placed among the top 10 worldwide threats to public health [9,10,11,12,13].

Most HAIs are caused by multidrug-resistant (MDR) strains of *A. baumannii*. The critical or priority 1 case of the World Health Organization (WHO) relating to the AMR problem belongs to carbapenem-resistant *A. baumannii* (CRAB) strains [5,9]. As previous studies show, the highest rate of multidrug-resistant (MDR) strains of *A. baumannii* belong to the Middle East and Europe, while the lowest rate of MDR strains of *A. baumannii* is associated with North America.

With the global increase of antimicrobial abuse or overuse, three groups of microbial-resistant strains, including *A. baumannii*-resistant strains, have appeared. These three categories are composed of MDR (resistant to the major portion of antimicrobials excluding two antimicrobials) strains, pandrug-resistant (PDR/resistant to all of the present antibiotics) strains and extensively drug-resistant (XDR/resistant to >three classes of antibiotics) strains [6].

In total, *A. baumannii* uses five resistant mechanisms, including the production of β-lactamase enzymes, the production of multidrug degrading enzymes, plasma membrane-related proteins’ molecular patterns’ alteration, recruitment of MDR pumps, and ribosomal methylation [6,14,15,16].

In recent years, the use of next-generation sequencing (NGS) technologies such as whole-genome sequencing (WGS) techniques has led to an effective increase in the knowledge concerning bacterial resistomes, molecular epidemiology typing and strain detection. The WGS technique enables us to screen very close strains of *A. baumannii* differing via a single nucleotide [5,9,17,18].

The genomic plasticity of the pathogenic strains of *A. baumannii* seems high; according to global reports, these pathogens possess versatile genomic sequences in different geographical areas worldwide [19,20].

The major goal of the present review literature is to highlight the main virulence factors, molecular pathogenesis and a genomic pool of CRAB strains.

## 2. Genomic Pool

The study of bacterial pan-genomics started about two decades ago, and the progression of advanced high-throughput technologies, e.g., NGS methodologies, has supported this effective discipline [21]. According to pan-genomic classical categorization, the bacterial genomic pool, including the *A. baumannii*’s genomic pool is composed of a core genome (>99%) (the softcore genome is a stable genomic pool which can be detected among 95–99% oe all strains), which contains those genes distributed among all related strains and are involved in cell division, energy production, genetic processes, metabolism, etc. In this regard, as a part of core genomic pool, the housekeeping genes are good examples because they are involved in different types of vital cellular activities, e.g., replication, gene expression, translation, etc.) [22,23,24,25,26,27,28,29,30]. The accessory genome genes (flexible, dispensable, and adaptive), which is distributed among ≥2 related strains (the subset of accessory shell genes; (detectable within 15–95% of the genomic datasets)) have contributed to adaption to niches, virulence factors and antibiotic resistance. The dispensable genes are normally acquired via HGT or an occurrence of paraphyletic evolution] [22,23,24,30,31,32], and singleton genes [unique, exclusive, cloud genes (detectable in <15% of the genomic datasets)], which are recognized in just one strain (singletons are normally acquired through the HGT feature and they are in association with effective adaptation, virulence and pathogenicity in pathogenic bacteria) [22,23,24,30,33,34,35].

The median length of *A. baumannii* genomes is about 3.96 Mb with a median DNA G+C content of 39% (https://www.ncbi.nlm.nih.gov/genome/?term=Acinetobacter+baumannii, accessed on 11 March 2024).

The bacterial pan-genomes can be determined to be closed or open. In this regard, by the recruitment of Heap’s formula (*n* = *κ* * *N*^−α^), in which *n* is the number of genes associated with a determined number of genomes, κ and α are recognized as free items that can be empirically obtained (α = 1 − γ), and *N* is the number of genomes. However, when α > 1, it means that a pan-genome is closed, and when α ≤ 1, it means that the pan-genome is open [22,36]. The pan-genomic structural studies show that *A. baumannii* possesses an open pan-genome bearing a large number of MGEs such as Insertion sequences (ISs), transposons (Tns) and integrons in genomic islands. Furthermore, *A. baumannii* may bear the resistance genes on a wide range of plasmids from 2 kb to >100 kb in length [37,38,39]. As the reported results indicate, the occurrence of recombination and a variety of gene exchange mechanisms may lead to the entrance of new genes into bacterial genomes like *A. baumannii* [6,40]. These facilities expand the genomic pool of the *A. baumannii* strains through the addition of new genes. In this manner, the novel genes are detectable in newly sequenced genomes pertaining to *A. baumannii*. The α parameter in Heap’s law regarding the pan-genome of *A. baumannii* equals 0.71 (0 < 0.71 < 1). This amount of α confirms that the pan-genome of *A. baumannii* should be open. This feature is correlated with universal high genomic plasticity among *A. baumannii* strains [22,31,36,41,42]. 

These characteristics of *A. baumannii* strains, including the presence of a wide range of different novel genes in different strains’ genomes of *A. baumannii*, which is the outcome of gene duplications or HGT via MGEs, have been shown by Rodrigues et al. [6]. As the recorded reports show, the ARGs are located in both the core and accessory genomic pool in *A. baumannii*. ISs, integrase and transposase enzymes, are effective means in genetic exchanges with other bacterial species and strains through HGT to add the ARGs to the *A. baumannii* accessory genomic pool. Hence, HGT contributes effectively to the bacterial evolution of pathogenicity and virulencity in *A. baumannii*. Furthermore, some plasmids were identified bearing integrated intact prophages in *A. baumannii*. These prophages may be integrated within the plasmids through direct integration or via recombination (with the bacterial chromosome). The acquired prophage-encoded genes in *A. baumannii* seem to be involved in bacterial MDR. The reported results depict effective participation of prophages in *A. baumannii’s* pathogenicity evolutionary pathway and its spread [43].

The ISs in *Acinetobacter* strains, including *A. baumannii* (ISAba), are key genetic elements for the activation and transmission of carbapenem-hydrolyzing β-lactamase enzymes. Indeed, the ISs are responsible for the global concern regarding *A. baumannii* infections in public health systems. ISs (1 kb) are known as the simplest and smallest MGEs. Therefore, ISAba elements may lead to significant genomic variability in *A. baumannii* strains [38,44,45,46,47,48].

As previous studies show, both ISs and Tns have the same structures and recruit transposases to be inserted within a genomic site. Tns transfer different ARGs [38,48,49]. Integrons and site-specific recombinases are other elements belonging to MGEs that contribute to acquiring resistance cassettes. The gene cassettes relate to integrons and can encode antimicrobial resistance. The gene cassettes act as small non-autonomous MGEs, normally composed of a promoter-free open reading frame (ORF) and a recombination site known as *attC* [38,50,51]. The miniature inverted-repeat transposable elements (MITEs) are another non-autonomous structure that is a member of MGEs. It seems that ancestral ISs and Tns are the main sources of these degraded elements. Although MITEs have no ORFs in most cases, their involvement in antibiotic resistance promotion among Acinetobacter strains is a global concern [38,52,53,54].

In addition, Rodrigues et al. [6] showed that the strains of *A. baumannii* possess considerable gene permutation in their genomes while they are not significantly clonal. This feature has been detected via the recognition of several gaps between the strains’ genomes through the bioinformatic genomic comparative in silico assays in bioinformatic (dry) labs [6,39,55,56,57]. Rodrigues et al. [6] detected a variety of genomic islands, including metabolic islands, pathogenicity islands (PAIs), resistance islands and symbiotic islands within the genomic pools of strains belonging to *A. baumannii*. The highest number of the islands were associated with resistance islands, while the lowest number was related to symbiotic islands. The identified resistance genes within the resistance islands pertaining to some enzymes and efflux pumps (Figure 1) [6,58,59]. Furthermore, *bla*_oxa_ genes (including a wide range of D class β-lactamase variants) with a high diversity of variants are detectable in pan-resistome studies in association with *A. baumannii* [12,60,61].

The outer membrane vesicles (OMVs) are effective vehicles for transferring DNA fragments among bacterial species and strains. In this regard, the plasmid-borne genes of *bla*_NDM-1_ and *bla*_OXA-23_ and/or free environmental DNA molecules ((e-DNAs), whether via bacterial cell lysis or type VI secretion system (T6SS)) can be transferred from an *A. baumannii* strain to another through the OMVs. In recent years, it has been reported that chromosomal ARGs can be transferred between *A. baumannii* strains via the phage-mediated HGT. Therefore, the transmission of ARGs, e.g., *bla*_NDM-1_, can occur via generalized phage transduction among the MDR strains of *A. baumannii* instead of direct cell–cell interaction. In addition, phages are suitable vehicles for transferring genomic DNA fragments bearing ARGs between MDR strains of *A. baumannii* [66]. As recent investigations show, prophages are significant cargo capable of encoding ARGs in their bacterial hosts (*A. baumannii* strains). This feature may support the occurrence of lysogenization as a pivotal feature in phage-mediated HGT [67]. The most frequent feature pertaining to colistin resistance in *A. baumannii* is the occurrence of mutations in chromosomal operon genes of *pmrCAB*. However, it has been most recently observed that the colistin resistance feature can be transduced even by the *A. baumannii* bacteriophage of Φ19606 through the transmission of the *eptA1* gene (the homolog of *pmrC* gene) via the HGT. Up until now, the *A. baumannii* bacteriophage of Φ19606 is the first phage-mediated colistin resistance feature in bacterial cells [62].

## 3. Virulence

### 3.1. Virulence Factors and Molecular Pathogenesis

Through a wide range of pan-genomic studies, it has been recognized that *A. baumannii* is an open pan-genomic bacterial pathogen; hence, *A. baumannii* encompasses a variety of virulence factors with different functions and activities. Because of the importance of these virulence factors, Table 1 shows a complete package of the identified virulence factors together with the related genes [3,68,69].

In accordance with the latest studies, the deciphered virulence and pathogenesis factors in *A. baumannii* are divided into nine groups, including capsular exopolysaccharide (CPS), efflux pumps, lipopolysaccharide (LPS), lipooligosaccharide (LOS), outer membrane proteins (OMPs), pili, metal ion uptake systems, two-component systems (TCSs) (virulence factors’’ regulators), and secretion systems (SSs) (Figure 1 and Figure 2) [68]. These items are summarized in Table 1.

In recent years, pan-genomic investigations indicated that the majority of virulence factors and ARGs are transmitted via different MGEs, including Tns, ISs, integrons, bacteriophages, etc. Many of these MGEs are transferred by a mass of plasmids through different types of HGT [66,67,70]. Table 2 indicates the related items.

### 3.2. Capsular Exopolysaccharide (CPS)

CPS is known as the main virulence factor arming *A. baumannii* stains. This structure acts as a superficial multifunctional virulence factor that pathogenic bacteria like *A. baumannii* use to survive in unfavorable environmental conditions, such as the presence of antimicrobials, human host cell immune defense mechanisms and prolonged desiccation (Table 2) [68,71,72]. The omission of CPS expression in CPS-producing *A. baumannii* strains may significantly affect bacterial virulency and pathogenesis. 

Indeed, CPS is recognized as a pivotal protection for bacterial survival in vivo (Figure 1 and Figure 2) [73]. CPS is composed of repeating K units; however, the K units differ in composition, length and structure (branched or unbranched). The results reported by analytical investigations show that K units are constructed by derivatives of simple UDP-linked sugars. These sugar molecules are mainly comprised of glucose and galactose and sometimes form complex sugars, e.g., non-2-ulosonic acid. This diversity among K units depicts the presence of effective variabilities (>100 different types) in capsular loci. Moreover, the structure of K units directly determines the virulence and pathogenesis of *A. baumannii* strains [71,72,74,75]. Those genes between *fkpA* and *lldP* genes encode proteins involved in capsular polysaccharides biosynthesis and/or transmission processes [74]. The two-component system (TCS) of BfmRS is involved in the upregulated expression of K loci and increased capsule secretion [71,76].

### 3.3. Efflux Pumps

Efflux pumps are important bacterial defensive mechanisms that support bacterial pathogens, e.g., *A. baumannii* strains, in the presence of antimicrobial drugs. The efflux pumps extrude the antimicrobials from the exposed bacteria. Although *A. baumannii* encompasses effective structures of outer membranes (OMs) with low permeability against the large-sized molecules of antimicrobial drugs, the efflux pumps have a vital role in this regard (Figure 1). Because of the wide varieties of efflux pumps in different prokaryotic cells of bacteria, these pumps are categorized into six different groups, including the ATP Binding Cassette (ABC), Small Multidrug Resistance (SMR), Major Facilitator Superfamily (MFS), Proteobacterial Antimicrobial Compound Efflux (PACE), Multidrug And Toxic compound Extrusion (MATE), and Resistance Nodulation Division (RND). The identified efflux pumps among the *A. baumannii* isolates are shown in Table 2 and Figure 2. Some members of RND may also be involved in biofilm formation [77,78,79,80]. The MFS group possesses the widest range of varieties and, therefore, is present in different types of life domains. The MFS transporters are classified into three categories called antiporters, symporters and uniporters [68,80,81].

In accordance with reported results from different studies, the efflux pumps pertaining to RND family members may be responsible for the appearance of intrinsic resistance among Gram-negative bacteria, including *A. baumannii* strains, in exposure to a wide range of antimicrobial groups [80,81,82]. AdeABC is known as the first member of the RND efflux pump, which was detected in clinical isolates of *A. baumannii* strains. The AdeABC contributes to bacterial resistance against various antimicrobial groups, e.g., β-lactams, chloramphenicol, aminoglycosides, fluoroquinolones, erythromycin, and so on. Moreover, this member of RND is increasingly produced among MDR strains isolated from clinical samples. Hence, AdeABC may be identified as the most clinical-related RND efflux pump [68,79,80,82,83]. The efflux pumps contribute to other vital bacterial activities rather than MDR characteristics, including virulence, surface-related motility, quorum sensing activities and the related pathways, e.g., biofilm formation (Figure 1 and Figure 2) [84,85].

**Figure 2 antibiotics-13-00257-f002:**
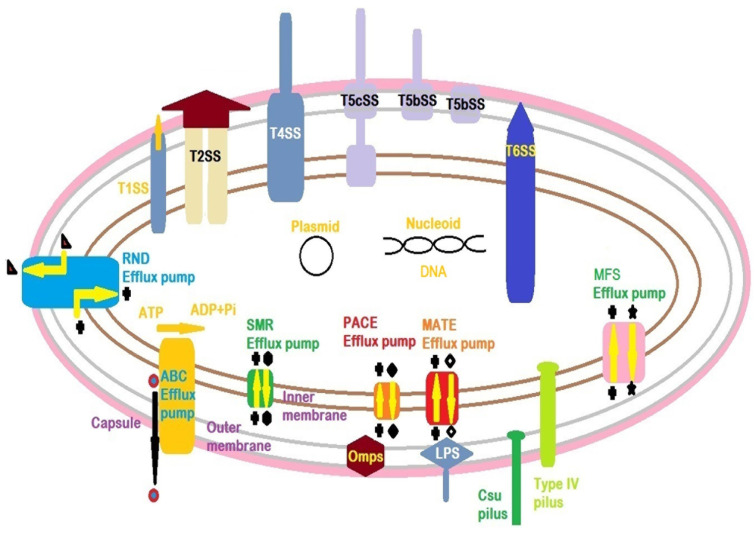
A schematic bacterial cell pertaining to *Acinetobacter baumannii*. The situation and structures of important virulence factors include the capsule, outer membrane proteins (Omps), lipopolysaccharide (LPS), Csu pilus, type IV pilus, efflux pumps and secretion systems (SSs). As shown, the secretion systems exert molecules (proteins) into the extracellular zone. Two types of T5bSS and T5cSS have been detected in *A. baumannii*. Although there are five sub-groups of T5SS (e.g., T5aSS, T5bSS, T5cSS, T5dSS and T5eSS) in Gram-negative bacteria, only two sub-groups of T5bSS and T5cSS are detected among the *A. baumannii* strains. Among these two secretion systems, T5cSS is the main secretion system that can be identified in *A. baumannii* populations. The efflux pumps—excluding ABC and RND families—take protons from the bacterial periplasm space into the cytoplasm and simultaneously excrete antibiotics from the bacterial cytoplasm into the periplasm space. The ABC family consumes ATP to excrete the antibiotic molecules from the bacterial periplasm space into the extracellular space. It acts as uniport efflux pump. On the other hand, the RND family takes the protons from the periplasm space into the cytoplasmic space and simultaneously excretes the antibiotic molecules from the periplasm space into the extracellular space [68,80,86].

### 3.4. Lipopolysaccharide (LPS)

LPS is the main molecule that forms the OM in Gram-negative bacteria. The structural integrity of the OM is covered by the LPS composition through their presence in high abundance. It also determines the permeability capabilities of bacterial OM in exposure to hydrophobic molecules. The LPS (smooth LPS) molecule is composed of three segments: lipid A (a glycolipid), a core (made up of oligosaccharide) and a repetitive O-antigen (an effective immunogen structure) (Figure 1 and Figure 2). The core bridges lipid A to the O-antigen. These properties of LPS make it a microbe-associated molecular pattern (MAMP) for human host innate and adaptive immune system components such as toll-like receptors (TLRs) [3,87,88,89,90,91,92,93]. TLRs and interleukins (ILs) are important immune biomolecules that can be induced by stranger molecules, such as different parts of microbial components. TLR activation may lead to induce TLR signaling pathways that results in activation of several immune cells and immune biomolecules in a cascade process. In this regard, the LPS of *A. baumannii* triggers the production of a cascade of biomolecules such as pro-inflammatory cytokines (e.g., IL-8, tumor necrosis factor-α (TNF-α), CCL4, TLR4, etc.) [3,87,92,93,94]. The lipid A, as a negatively charged component of the LPS biomolecule, makes the bacterial cells of *A. baumannii* susceptible to the cationic polypeptide of colistin. The *lpxABCD* gene operon encodes the LPS components (Table 2) [68,95].

### 3.5. Lipooligosaccharide (LOS)

LOS, which is known as rough LPS, misses the immunogenic component of the O-antigen within its structure (Table 2) [91]. The high-variated *oc* locus is usually identified in Gram-negative bacteria, e.g., *A. baumannii*. The *oc* locus is located in the region between *ilvE* and *aspS* genes. The early detection of OCL1–3 variants was achieved in a genomic pool of *A. baumannii*. The biomolecule of LOS contributes to a wide range of activities, including induction of pro-inflammatory immune responses (as aforementioned in association with LPS), bacterial motility, killing-resistant activity against opsono-phagocytosis, superficial attachment, bacterial susceptibility in exposure to colistin as a polymyxin antimicrobial agent and resistance to serum antimicrobial peptides in the human body [95,96,97,98,99,100,101,102,103,104,105,106,107].

### 3.6. Outer Membrane Proteins (OMPs)

OMPs are important bacterial virulence factors in *A. baumannii* strains for bacterial adherence in the human host cells (Table 1 and Table 2, Figure 2). OmpA contributes to bacterial adhesion (*A. baumannii* ATCC17978 and ATCC19606) to the epithelial cells [69]. OmpA is capable of attaching to human host integrins, junctional adherence proteins, fibronectin and cytoskeleton. In the AB5075 strain of *A. baumannii*, the OmpA binds to desmosomes and hemidesmosomes to modulate the processes of cell-to-cell contact in epithelial cells and weakening cell-to-cell interactions. This strategy helps the bacterial cells of *A. baumannii* to invade the epithelial cells of the host body. Therefore, the overexpression of OMPs in *A. baumannii* is presumed to be an effective means for bacterial adhesion to human host cells as a part of bacterial virulence and pathogenesis [69,108,109].

Although OMPs in Gram-negative bacteria are located within the bacterial OM, they participate in different types of bacterial activities, including bacterial adhesion and invasion, biofilm formation, occurrence of antibiotic resistance to a wide range of antibiotic families, host-mediated stresses, bacterial survival and induction of apoptosis (programmed cell death) through the activation of caspase enzymes within the human host cells [57,110,111,112,113]. In addition to OmpA, OmpW, Omp33–36, outer membrane carboxylate channels (Occ) and Carbapenem susceptible protein (CarO) are the pivotal OMPs involved in bacterial virulence and pathogenesis of *A. baumannii* (Table 2) [68].

The highest portion of porins in *A. baumannii* belongs to OmpA. This porin is composed of two domains: an N-terminal (made up of an eight-stranded antiparallel β-barrel) situated within the OM and a C-terminal with a globular configuration interacting with the bacterial cell wall. OmpA induces TLR2 in epithelial cells of the lung human host and promotes the permeability of the lung epithelial cells [110,111,114,115,116,117]. A major portion of OmpA can be transferred into the human host cells through the OMVs, produced by the bacterial cells of *A. baumannii*. The OmpA molecules enter the human host cells through the fusion process of OMVs [118].

OmpW encompasses an eight-stranded antiparallel β-barrel configuration located in OM. It participates in bacterial adhesion, invasion, biofilm formation and iron acquisition. As the reports show, the TCS of BfmRS regulates the expression of the OmpW encoding gene and those genes that contribute to the expression of siderophores and oxidative osmotic stress response [119,120,121,122].

CarO is another member of OMPs involved in the carbapenem resistance feature and acquisition of some amino acids, e.g., glycine and ornithine, in *A. baumannii* strains (Table 1). As previously mentioned, CarO contributes to bacterial adhesion, dissemination and invasion into the human host cells, such as lung epithelial cells [123,124,125].

Occ proteins form channels with narrow pore sizes. Occ proteins are structured as monomeric 18-stranded β-barrels, participating in small molecule transmission. Five orthologs of these proteins, including OccAB1, OccAB2, OccAB3, OccAB4 and OccAB5, have been detected in *A. baumannii* ATCC17978. OccAB1 or former OprD belongs to Occ family proteins in *Pseudomonas aeruginosa*. Although OprD contributes to carbapenems acquisition in bacterial cells of *P. aeruginosa*, OccAB1 has no relationship with carbapenem acquisition in bacterial cells of *A. baumannii*. Recently, a new homolog of OprD has been detected in clinical isolates of *A. baumannii*, which is related to hypervirulence in CRAB strains [13,126,127,128,129,130].

The Omp33–36 protein in *A. baumannii* strains is involved in bacterial adhesion, invasion, cytotoxicity and resistance to carbapenems. In contrast to OmpA, which encompasses nuclear localization signals to induce the apoptosis feature in host cells, the Omp33–36 protein does not express this feature. However, the internalized Omp33–36 is able to activate the caspase enzymes to induce apoptosis feature in host cells [131].

### 3.7. Pili

Pili are known as pivotal structures in pathogenic bacteria for bacterial adherence to biotic (human host cells) or abiotic (catheter) surfaces to have effective pathogenesis and infection (Table 1). Indeed, these superficial, short, antigenic thread-like appendages are detectable in both Gram-negative and Gram-positive bacteria categories. Because of the high diversity of pili in Gram-negative bacteria, they have been categorized into five different groups: chaperone-usher pili (CUP), curli fimbriae, type IV pili, type V pili, and conjugative type IV secretion pili (Table 2). The CUP pili are effective virulence factors in clinical isolates of *A. baumannii* strains because of their capability to adhere to biotic and abiotic surfaces in human patients with different infectious diseases like urinary tract infections [69,132,133]. The immunogenic Csu pili (Figure 2) belongs to type I CUP and has been detected in *A. baumannii* strains. These pili are involved in bacterial attachment to hydrophobic (tight adhesion) and hydrophilic (very loose adhesion) surfaces but not to the epithelial cell line surfaces [84,132,134,135].

On the other hand, type IV pili in the bacterial cells of *A. baumannii* are involved in twitching, superficial adhesion, biofilm formation and natural transformation (Figure 2). These pili activities are regulated by the GacSA TCS. The PilA variants have different activities in oppose to surface chemistry [136,137,138,139,140]. Curli fibers (Table 2) are constructed by different types of Csg pilins. Although curli fimbriae are detected in some strains of *E. coli*, no report has been recorded in association with a complete machinery system of curli biosynthesis in *A. baumannii* [132,141].

### 3.8. Metal Ion Uptake Systems

Metal ion uptake systems are vital means for pathogenic bacteria like *A. baumannii* to survive (Table 1). In this regard, metal ions associated with iron, zinc and manganese are essential elements for eukaryotic host cells, e.g., human host cells and microbial pathogens such as *A. baumannii*. On the one hand, iron, as an important metal ion, is restricted through employing associated proteins, including hemoglobin (the major protein), ferritin, lactoferrin, transferrin and calprotectin (a metal-chelating antimicrobial protein pertaining to the innate immune system) by the human host cells. On the other hand, pathogenic bacteria like *A. baumannii* have their own iron acquisition systems, e.g., siderophore proteins (e.g., three classes of acinetobactin, baumannoferrin, and fimsbactin) and heme acquisition systems (Table 2) [142,143]. The enzyme phospholipase C, produced by bacterial cells of *A. baumannii*, disrupts the human host red blood cells (RBCs) to uptake the released hemoglobin. Then, the bacterial heme acquisition system converts the released form of the iron ion into the requisite form for bacterial cells. The bacterial heme acquisition system is a multi-sectional and multi-functional system. In this regard, the heme oxygenase component is responsible for converting ferrous ions (Fe^3+^) into ferric ions (Fe^2+^) [144,145,146,147].

Siderophore activity needs energy produced via the proton motive force (pmf) situated across the bacterial inner membrane. Furthermore, this process is modulated by a protein complex composed of TonB-ExbB-ExbD and ABC transporters. The expressed siderocalin molecules by neutrophils prevent the iron acquisition activity of the bacterial siderophores [68,76,142,147]. As published reports show, the evolution of iron acquisition systems in *A. baumannii* can be occurred through different genetic exchanges, e.g., HGT [148]. Zinc is another important metal essential for both human host and bacterial pathogens, e.g., *A. baumannii*. Bacterial enzymes like metalloproteins need zinc for their activity. The overexpression of *znuABC* genes by *A. baumannii* to acquire further zinc ions from the environment is modulated by the zinc uptake regulator (Zur), a zinc-binding protein about the ferric uptake regulator (Fur) protein family [149,150,151,152].

Metal ions of Manganese (Mn^2+^) play a key role as co-factors in a wide range of enzymes, such as superoxide dismutase, which contributes to protecting bacterial cells of *A. baumannii* from reactive oxygen species (ROS) in the feature of oxidative stress. *A. baumannii* strains recruit natural resistance-associated macrophage protein (NRAMP) family proteins like MumT to cover themselves against the human host immunoprotein of calprotectin. MumT is a transporter enzyme that participates in manganese and urea metabolisms [153,154]. As the reported results show, a pathogen like *A. baumannii* needs a rigorous regulatory mechanism for its metal uptake systems and homeostasis for successful virulence and pathogenesis within the human host body [76].

### 3.9. Two-Component Systems (TCSs)

TCSs are pivotal systems that control the phenotypic switching processes. Due to this fact, these signal transduction systems can be detected in both archaea and bacteria. The importance of TCSs lies in their contribution to bacterial adaptation and sensing in exposure to different environmental factors and situations. A TCS is normally comprised of two components: a cytoplasmic membrane-bound sensor (e.g., a histidine kinase) and a response regulator, which is involved in DNA-binding and transcription processes. It seems that a bacterial genome may bear between 50 and 60 genes that encode TCS proteins. According to previous reports, *A. baumannii* strains encompass up to 20 TCSs with different functions. TCSs are able to change the bacterial genetic reactions in exposure to environmental factors, e.g., antibiotic resistance, virulency and pathogenesis (Table 2) [68,76,77,155,156].

### 3.10. Secretion Systems (SSs)

SSs are known as the major virulence factors because they can transfer bacterial virulence factors to the outside of the bacterial cells. This strategy induces the human host immune system and may support the virulencity and pathogenesis of pathogens like *A. baumannii*. Until now, six SSs—including T1SS–T6SS—have been discovered in Gram-negative bacteria such as *A. baumannii*. These SSs have their own compositions, functional activities, structures and roles in bacterial virulence, pathogenesis and antimicrobial resistance (Table 2, Figure 1 and Figure 2) [86].

**Table 2 antibiotics-13-00257-t002:** General microbial virulence factors arsenal detected in *A. baumannii*.

Structures		Virulence Genes	Gene Position	Virulence Factors	Roles	References
Capsule (CPS)		>100 unique capsule loci (*KL*) with different sizes in length (between 20 and 35 kilobases (kb))	Between the *fkpA* and *lldP* genes on the chromosome	Capsular biosynthesis and export	Bacterial pathogenicity, Virulence, Antimicrobial resistance, Persistence, Evasion of host immune system (antiphagocytosis), reduction of interactions between human host and pathogen	[3,64,68,71,74,157]
Efflux pumps	ATP binding cassette (ABC) transporter	*A1S-0536*	Chromosome	A1S-0536	Resistance to erythromycin	[3,68,79,80]
		*A1S-1535*		A1S-1535	Resistance to chloramphenicol and gentamicin	
		*abuO*		AbuO	Response to oxidative stress	
		*macAB-tolC*		MacAB-TolC	Potentially resistance to macrolides and tigecycline	
	Multidrug and toxic compound extrusion (MATE)	*abeM*		AbeM	Resistance to fluoroquinolones and disinfectants	[3,6,68,79,80]
	Major facilitator superfamily (MFS)	*abaF*	Chromosomal genomic islands	AbaF	Resistance to fosfomycin	[3,68,79,80,158]
		*abaQ*		AbaQ	Resistance to quinolones	
		*amvA*		AmvA	Resistance to erythromycin	
		*cmlA*		CmlA	Resistance to chloramphenicol	
		*craA*		CraA	Resistance to chloramphenicol	
		*emrAB*		EmrAB	Resistance to colistin and adaptation to osmotic stress	
		*tetA*	Plasmids and MGEs	TetA	Resistance to tetracycline and tigecycline	
		*tetB*		TetB	Resistance to minocycline and tetracycline	
	Resistance nodulation division (RND)	*abeD*	Chromosome	AbeD	Resistance to benzalkonium chloride, ceftriaxone, gentamicin, rifampin, tobramycin, killing the host cells	[3,6,65,74,75,76,77]
		*acrAB*		AcrAB	Resistance to disinfectants, colistin and tobramycin	
		*adeABC*		AdeABC	Resistance to aminoglycosides, chloramphenicol, fluoroquinolones, pentamide, tetracyclines, trimethoprim and osmotic stress	
		*adeDE*		AdeDE	Resistance to chloramphenicol, erythromycin, tetracycline, amikacin, meropenem, ceftazidime, rifampin and ciprofloxacin	
		*adeFGH*		AdeFGH	Resistance to clindamycin, chloramphenicol, fluoroquinolones, tetracycline-tigecycline and trimethoprim, biofilm formation	
		*adeIJK*		AdeIJK	Resistance to erythromycin, β-lactam antibiotics, trimethoprim, tetracycline, fusidic acid, chloramphenicol, novobiocin, lincosamides and rifampin	
		*adeL*		AdeL	Resistance to fluoroquinolones and tetracycline	
		*adeN*		AdeN	Resistance to macrolides, tetracycline, cephalosporins, carbapenem, penems, fluoroquinolones and rifamycin	
		*adeXYZ*		AdeXYZ	Similar to AdeIJK in phenotypic, structural and genetic characteristics (homologs)	
		*arpAB*		ArpAB	Resistance to tobramycin and amikacin	
		*czcABCD*		CzcABCD	Resistance to heavy metals, e.g., copper	
	Proteobacterial antimicrobial compound efflux (PACE)	*A1S-1503*	Chromosome	A1S-1503	Resistance to disinfectants	[3,68,79,80]
		*aceI*		AceI	Resistance to disinfectants	
	Small multidrug resistance (SMR)	*abeS*	Chromosome	AbeS	Resistance to novobiocin, erythromycin, chloramphenicol, fluoroquinolones and disinfectants	[3,68,79,80]
		*qacE*		QacE	Resistance to disinfectants	
Lipopolysaccharide (LPS)		*lpxACD* operon; *lpxB*	Chromosome, plasmids	lipidA biosynthesis, LPS biosynthesis	Bacterial surface-associated motility, Microbe associated molecular pattern (MAMP), Immune system activation through triggering the expression of versatile pro-inflammatory cytokines, e.g., toll-like receptor 4 (TLR4), interleukin 8 (IL-8), Tumor necrosis factor-α (TNF-α) and CCL4, Bacterial susceptibility against colistin	[62,87,94,98,159,160]
Lipooligosaccharide (LOS)		Outer core (*OC*) loci (*OCL*)	Between *ilvE* and *aspS* genes on the chromosome	lipid-carbohydrate surface structure	Antimicrobial peptides resistance, Bacterial adhesion, Bacterial resistance against human host opsonophagocytotic activities, bacterial cell motility, Induction of expression of several pro-inflammatory cytokines	[3,96]
Outer membrane proteins (OMPs)		*carO*	Chromosome, plasmids	CarO	Resistance to Carbapenems, uptake of glycine, imipenem, and ornithine, contribution to bacterial adhesion, invasion and dissemination	[3,68,77,119,125,161]
		*occAB1*–*AB5*		OccAB1–AB5	Substrates translocation, metal ions acquisition, e.g., iron (Fe^2+^) and magnesium (Mg^2+^), antibiotics (β-lactams) and amino acids uptake, participation in nutritional immunity and stress survival caused by the host (host–pathogen interactions)	
		*omp33*–*36*		Omp33–36	Resistance to carbapenems, activation of caspase enzymes of 3 and 9 and apoptosis within the host cells, cytotoxicity, bacterial adhesion and invasion into the host’s epithelial cells	
		*ompA*		OmpA	Resistance to β-lactams and colistin, iron siderophores (e.g., acenitobactin) acquisition, cytotoxicity, bacterial adhesion through fibronectin (irreversible attachment), invasion and persistence, induction of reactive oxygen species (ROS) and apoptosis within the host cells, it has been recognized in up to 81% of isolated strains of *A. baumannii*	
		*ompW*		OmpW	Bacterial adhesion and invasion into pulmonary epithelial cell lines, cytotoxicity, iron acquisition	
Pili	Chaperon-usher type I pili	*csuA/BABCDE*	Plasmid/chromosome	Formation of chaperone-usher Csu fimbriae; CsuA/B (Shaft of the pili (Major subunit)), CsuA (Minor subunit), CsuB (Minor subunit), CsuC (Chaperon), CsuD (Usher), CsuE (Adhesin tip)	Biofilm formation on abiotic surfaces, irreversible attachment; recognized in up to 100% of the isolated strains belonging to *A. baumannii*	[68,77,162,163]
	Type IV pili	*pilApgyA*, *pilBCD*, *pilTU*	Plasmid/chromosome	Formation of type IV pili; PilA (Major subunit), PgyA (O-glycosylase), PilB (putative traffic ATPase), PilC (putative inner membrane platform protein), PilD (putative prepilin peptidase), PilT (putative retraction ATPase), PilU (putative retraction ATPase)	Biofilm formation, host-cell adhesion, twitching motility, HGT, microcolony formation	[77,139,164]
	Curli fiber	*csgBAC* *csgDEFG*		Amyloid protein (composed of major subunits of csgA)	Adherence, matrix formation, biofilm maturation; recognized in up to 70% of the isolated strains belonging to *A. baumannii*	[77,165]
	Type I fimbriae	*fimBEAICDFGH*	Chromosome	FimH (adhesin)	Bacterial cell adhesion (irreversible attachment), recognized in up to 50% of the isolated strains belonging to *A. baumannii*	[77,165]
	P fimbriae	*papIBAHCDJKEFG*	PAIs/chromosome	PapG (adhesin)	Biofilm formation (homologous to *Escherichia coli*); recognized in up to 80% of the isolated strains belonging to *A. baumannii*	[77,165]
Metal ion uptake systems	Phospholipase	*plc1*, *plc2*	Plasmid/chromosome	Phospholipase C	Red blood cell lytic and hemoglobin releasing enzyme, iron uptake and lipolytic activity	[3,68]
		*pld1*, *pld2*, *pld3*		Phospholipase D	iron uptake and lipolytic activity	[3]
	Acinetobactin (including three gene clusters of A1S-2392-A1S_2372)	*basAB*, *basCD*, *basFG*, *basHIJ*	Plasmid/chromosome	BasA–J	Biosynthesis of acinetobactin for iron uptake, persistence of the infection within the epithelial cell and apoptosis	[3,68,147,148,153,166]
		*barA*, *barB*		BarAB	Members of siderophore efflux system of the ABC superfamily, which secrete the produced acinetobactin via *bas* gene clusters	
		*bauABECD*, *bauF*		BauA–F	Receptor for complexes of ferric-acinetobactin to translocate these complexes into the bacterial cell of *A. baumannii*, persistence of the infection within the epithelial cell and apoptosis	
	Baumanoferrin (including one gene cluster of A1S_1647-A1S-2372)	*bfnABCDEFGHIJKL*		BfnA–L	Biosynthesis, translocation and the uptake of iron	
	Fimsbactins (including two gene clusters of A1S-2582-A1S_2562)	*fbsABCDEFGHIJKLMNOPQ*		FbsA–Q	Biosynthesis, translocation and the uptake of iron	
	mum operon	*mumRTLUHC*		MumR–C	MumR contributes to oxidative stress resistance and regulating of Mn homeostasis; MumT participates in Mn chelating, it also acts as Mn and urea transporter	
	Fur	*fur*		Fur	Transcriptional regulator of iron metabolism	
	Zinc uptake system	*zigA*		ZigA	Zinc homeostasis,	
		*znuBC*, *znuA*, *znuD*, *znuD2*		ZnuB (inner membrane channel), ZnuC (ATPase), ZnuA (periplasmic binding protein), ZnuD (outer membrane channel), ZnuD2 (outer membrane channel)	Zinc acquisition, homologous to ZnuABC system in *E. coli*, pathogenesis	
		*zur*		Zur	Transcriptional regulator of zinc metabolism	
Two-component systems (TCSs)	A1S_2811	*A1S_2811*	Plasmid/chromosome	A1S_2811	Hybrid sensor kinase, involved in motility (via flagella or pili), contribution to biofilm formation, regulation of quorum sensing	[155]
	AdeRS	*adeRS*		AdeR, AdeS	Involved in antibiotic susceptibility in *A. baumannii*, contribution to controlling the expression of ~600 genes (involved in, e.g., biofilm formation, multidrug-efflux activity (such as AdeABC) and virulence), directly or indirectly.	[76,155]
	BaeSR	*baeSR*		BaeS, BaeR	There is a homology between this TCS in *A. baumannii* and *E. coli*, triggered by sucrose, regulation of overlapping regulons relating to other present TCSs in *A. baumannii*, BaeSR may occur cross-talk with the other members of TCSs; it regulates the expression of AdeABC, ADEIJK, MacAB-TolC drug efflux pumps, Susceptible to tannic acid, contributes to bacterial antibiotic resistance	[76,155]
	BfmRS	*bfmRS*		BfmR, BfmS	This TCS (recognized in up to 92% of isolated strains of *A. baumannii*) is a type of sensor kinase that regulates the expression of *csu* operon (pili) in *A. baumannii*, contribution to biofilm formation (irreversible attachment), regulation on capsule production through controlling the exopolysaccharide biosynthesis (expression of K locus), regulation of bacterial TCSs and virulence in *A. baumannii*, BfmS phosphorylates the BfmR	[76,77,155]
	GacSA	*gacSA*	Transposon/chromosome	GacS, GacA	Regulation of ~680 genes in association with motility, pili formation, biofilm formation, bacterial resistance towards human serum, immune evasion, catabolism of aromatic compounds (paa operon, which has homology to *E. coli*), bacterial pathogenesis	[76,155]
	PmrAB	*pmrAB*	Plasmid/chromosome	PmrA, PmrB	Lipid A modification, contribution to colistin and polymyxin B resistance (via gene mutations)	[155]
Secretion system	T1SS	*tolC-hlyB-hlyD* gene cluster (homologous to *E. coli*)	Chromosome	TolC (a trimeric outer membrane protein interacting with HlyD), HlyB (an ATP-binding cassette transporter that provides the required energy), HylD (a periplasmic adaptor)	T1SS has cross-talk with T2SS and T6SS, Contribution to virulence, secretion of putative effectors, e.g., RTX-serralysin-like toxin and Bap and other effectors such as glycosidases, proteases, phosphatases and invasins which are involved in bacterial attachment, invasion, biofilm formation and pathogenesis	[68,86,167]
	T2SS	*gspC-M*, *pilD*	*gsp* genes dispersed into five clusters within the bacterial genome	GspC (a subunit of the inner membrane (IM) platform), GspD (an outer membrane (OM) complex), GspE (cytoplasmic ATPase), GspF (a subunit of the inner membrane (IM) platform), GspG-K (subunits of the periplasmic pseudopilus (GspG is known as major pseudopilin, while the others are recognized as minor pseudopilins)), GspL (a subunit of the inner membrane (IM) platform), GspM (a subunit of the inner membrane (IM) platform), PilD contributes to processes of cleavage and methylation	Secretion of enzymes and toxins such as CpaA protease, intimin-invasin lipoprotein of InvL and lipases of LipA, LipAN and LipH, pathogenesis, antibiotic resistance (e.g., resistance against ciprofloxacin)	
	T4SS (type F)	*traA-I*, *traK-N*, *traU-W*, *trbC*, *finO*	Plasmid	TraA (constitutes the extracellular section of the pilus), TraB (constitutes the IM platform of the T4SS), TraC (the cytoplasmic subunit of the T4SS), TraD (the cytoplasmic subunit of the T4SS), TraE, TraF (constitutes the IM platform of the T4SS), TraG (constitutes the IM platform of the T4SS), TraH (constitutes the OM core complex of the T4SS), TraI (contributes to nick), TraK (constitutes the OM core complex of the T4SS), TraL, TraM (participates in translocation initiation), TraN (constitutes the OM core complex of the T4SS), TraU (constitutes the IM platform of the T4SS), TraV (constitutes the OM core complex of the T4SS), TraW (constitutes the IM platform of the T4SS), TrbC (constitutes the IM platform of the T4SS), FinO (regulator)	T4SS is involved in DNA exchanges, e.g., HGT features such as transformation, conjugation which may lead to translocation of virulence genes, drug-resistance genes among bacterial cells, contribution to bacterial pathogenesis, colonization and proliferation within the eukaryotic host cells	[3,68,86]
	T5bSS	*cdiA*, *cdiB*, *abfhaB*, *abfhaC*	Chromosome	CdiA (toxin), CdiB (OM transporter)	Lethal proteins	[3,68,86]
				AbFhaB, AbFhaC	Adhesion (via fibronectin) to host cell, virulence, bacterial survival, biofilm formation	
	T5cSS The most popular T5SS in *A. baumannii*	*ata*	Chromosome	Ata	Autotransporter, biofilm formation, attachment to laminin and different types of collagens including I, III, IV and V, bacterial survival, bacterial invasion to host cells, induction of apoptosis (programmed cell death) process within the host cells	[3,68,86,112]
	T6SS	*asaA-tssBC-hcp(tssD)-tssEFG-asaB-tssM-tagFN-asaC-tssHAKL-asaDE* *vgrG-paar*	Plasmid/chromosome	TssA–M forms the core, and TagD–L constructs the accessory proteins of the T6SS structure TssL–M and TssJ form the membrane complex of the T6SS within the periplasmic space; TssE–G, TssK, VgrG and PAAR construct the cytoplasmic baseplate; TssA–C and Hcp (TssD) form the tail tube or sheath complex of the bacterial T6SS in *A. baumannii*	Bacterial virulence factor, pathogenicity factor in eukaryotic host cells, bacterial invasion and adhesion, antibiotic resistance	[86,168,169]

## 4. Conclusions

*A. baumannii* possesses a wide range of virulence factors with versatile abilities. Simultaneously, this pathogenic bacterium is supported by a significantly high plasticity in its pan-genome. In addition, *A. baumannii* exploits the HGT as an effective genetic exchange to acquire or transfer MGEs. These data and information reveal that despite the presence of a huge volume of pathogenicity in *A. baumannii*, we can also utilize this bacterial arsenal to protect ourselves against its global concern associated with public health systems worldwide. In this regard, we can find the related bacterial strategies and mechanisms to acquire different types of virulence factors and ARGs from different sources. By knowing these strategies and related resources, we can protect ourselves against their pathogenesis, dissemination, proliferation, antibiotic resistance, and transmission. Moreover, deciphering the different roles of bacterial cells and their components in induction of different immune system signaling pathways, mediators, modulators in human and non-human hosts provides us an opportunity to promote our capability against a versatile of infections caused by *A. baumannii* through a wide range of alternatives like new strategies in pharmaceutical therapies, immunotherapies and vaccine production.

## Figures and Tables

**Figure 1 antibiotics-13-00257-f001:**
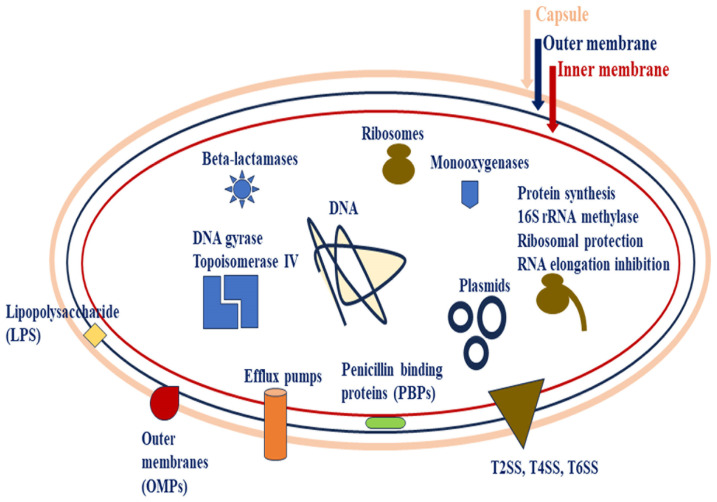
The main antibiotic resistance mechanisms can be divided into three categories in *A. baumannii*. In this regard, LPS modification or mutation in LPS bisynthesis genes, efflux pumps (promoted efflux), outer membrane proteins (OMPs/porins) (decrease in OMPs’ permeability) and secretion systems (T2SS, T4SS and T6SS) contribute to antibiotic resistance through transportation via bacterial cell membranes; antibiotic inactivation via enzymes (e.g., monooxygenase and β-lactamase enzymes); alterations and modifications in antibiotic target sites, such as mutations in antibiotic target enzymes (including DNA gyrase (single mutations in *parC* (topoisomerase IV encoding gene)) and/or in *gyrA* (DNA topoisomerase IV encoding gene) against fluoroquinolones), mutations in Penicillin-binding proteins (PBPs)), ribosomal protection proteins (RPPs) in the occurrence of non-covalent modifications of bacterial ribosomes against antibiotics (e.g., tetracycline), production of ribosome methylase enzymes (via plasmid-transposon-borne genes) against lincosamides, and 16S rRNA methylase enzymes against aminoglycosides. Plasmids are important MGEs directly participating in antibiotic-resistant genes via HGTs [3,62,63,64,65].

**Table 1 antibiotics-13-00257-t001:** Summarized information regarding the general microbial virulence factors arsenal detected in *A. baumannii*.

Virulence Factors	Functions
Capsule (CPS)	Antimicrobial resistance, host–pathogen interactions and pathogenesis
Efflux pumps	Antimicrobial resistance, biofilm formation, disinfectant resistance, heavy metal resistance and response to oxidative and osmotic stresses
Lipopolysaccharide (LPS)	Antimicrobial resistance, bacterial motility and induction of pro-inflammatory cytokines in humans
Lipooligosaccharide (LOS)	Antimicrobial resistance, bacterial adhesion, resistance to human opsono-phagocytosis and induction of pro-inflammatory cytokines in humans
Outer membrane proteins (OMPs)	Antimicrobial resistance, induction of apoptosis in human cells, bacterial adhesion, invasion and dissemination, cytotoxicity in human cells, host–pathogen interactions, induction of reactive oxygen species (ROS) in human cells, metal ion acquisition and substrate translocation
Pili	Bacterial adhesion, bacterial attachment (irreversible), biofilm formation, genetic exchange through HGT and motility (e.g., twitching)
Metal ion uptake systems	Bacterial metal (e.g., Zn, Fe, Mn) uptake, translocation, metabolism homeostasis, biofilm formation, quorum sensing regulation and response to oxidative stress
Two-component systems (TCSs)	Antimicrobial resistance, aromatic compound metabolism, bacterial pathogenesis, biofilm formation, capsule expression, regulation of bacterial motility, lipid A modification, regulation of some groups of efflux pumps and resistance to human serum
Secretion systems (SSs)	Antimicrobial resistance, bacterial adhesion, biofilm formation, bacterial colonization and proliferation within the human cells, bacterial survival, enzyme and toxin secretion, induction of apoptosis in human cells, genetic elements exchange via HGT and pathogenesis

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
