# Peer review of "Pan-Genome Plasticity and Virulence Factors: A Natural Treasure Trove for Acinetobacter baumannii"

_antibiotics, 2024, doi:10.3390/antibiotics13030257_

Round 1
Reviewer 1 Report
Comments and Suggestions for Authors
Summary
This is a timely review of virulence factors and genomics in A. baumannii. Contrary to the antibiotic resistance theme, there are not that many reviews that link virulence and genomics. In this regard, the review is very welcome. However, some issues should be addressed before this manuscript can be considered for publication.
Major comments
- A. baumannii has also been reported in several animals and plants, see references below, and the idea this bacterium can be a One Health issue has been put forward. The authors may want to add a couple of sentences about this in the introduction.
https://pubmed.ncbi.nlm.nih.gov/25616788/
https://pubmed.ncbi.nlm.nih.gov/37625430/
- Regarding the highly dynamic genome of A. baumannii, recent studies have shown that phages also play a role in horizontal gene transfer between different lineages of A. baumannii. Specially considering antibiotic and virulence genes
https://pubmed.ncbi.nlm.nih.gov/30337588/
https://pubmed.ncbi.nlm.nih.gov/31138576/
https://pubmed.ncbi.nlm.nih.gov/32109174/
https://pubmed.ncbi.nlm.nih.gov/38301486/
- As far as the 3rd and last section (“Virulence factors and molecular pathogenesis”) the authors want to mention which mobile genetic elements are transferring (if so) which virulence traits.
Minor comments
Lines 20-21: what do the authors mean by “bacterial ghost”?
Line 51: “the top 10 threats” reads better.
Line 121-123: please rephrase; “assays in dry labs” is rather ambiguous
Lines 310-314: most of the sentences are in italics and with a bigger font
Lines 356-357: How can we use A. Baumannii’s arsenal in our favour? Please elaborate.
Comments on the Quality of English LanguageThe manuscript is very readable but the English can be improved here and there.
Author Response
Dear Reviewer
First of all, we appreciate your positive and constructive comments.
Summary
This is a timely review of virulence factors and genomics in A. baumannii. Contrary to the antibiotic resistance theme, there are not that many reviews that link virulence and genomics. In this regard, the review is very welcome. However, some issues should be addressed before this manuscript can be considered for publication.
Major comments
- A. baumannii has also been reported in several animals and plants, see references below, and the idea this bacterium can be a One Health issue has been put forward. The authors may want to add a couple of sentences about this in the introduction.
https://pubmed.ncbi.nlm.nih.gov/25616788/
https://pubmed.ncbi.nlm.nih.gov/37625430/
Thank you very much for your effective comments. Your comments together with the references have been considered in our manuscript (page 1, lines 41-45).
- Regarding the highly dynamic genome of A. baumannii, recent studies have shown that phages also play a role in horizontal gene transfer between different lineages of A. baumannii. Specially considering antibiotic and virulence genes
https://pubmed.ncbi.nlm.nih.gov/30337588/
https://pubmed.ncbi.nlm.nih.gov/31138576/
https://pubmed.ncbi.nlm.nih.gov/32109174/
https://pubmed.ncbi.nlm.nih.gov/38301486/
Thank you very much for your effective comments. Your comments together with the references have been considered in our manuscript (page 3, lines 124-134; Page 4, lines 164-182).
- As far as the 3rd and last section (“Virulence factors and molecular pathogenesis”) the authors want to mention which mobile genetic elements are transferring (if so) which virulence traits.
Minor comments
Lines 20-21: what do the authors mean by “bacterial ghost”?
Thank you very much for your effective comments. Your comments have been considered in our manuscript and changed to "bacterial monster"
Line 51: “the top 10 threats” reads better.
Thank you very much for your effective comments. Your comments have been considered in our manuscript.
Line 121-123: please rephrase; “assays in dry labs” is rather ambiguous
Thank you very much for your effective comments. Your comments have been considered in our manuscript.
Lines 310-314: most of the sentences are in italics and with a bigger font
Thank you very much for your effective comments. Your comments have been considered in our manuscript.
Lines 356-357: How can we use A. Baumannii’s arsenal in our favour? Please elaborate.
Thank you very much for your effective comments. Your comments have been considered in our manuscript (Pages 25-26; lines 459-468).
Comments on the Quality of English LanguageThe manuscript is very readable but the English can be improved here and there.
Thank you very much for your effective comments. Your comments have been considered in our manuscript.
Reviewer 2 Report
Comments and Suggestions for Authors
This is an excellent review on resistance mechanisms of Acinetobacter.
The manuscript is comprehensive and informative.
The organization is poor. The items are so long that the reader could get lost. -I suggest more titles under 3. virulence
-Also suggest a simple figure to show all the mechanisms of resistance with their commonality and subtypes.
-Figure 1 is excellent but not comprehensive and does not help more than showing the location of the efflux pumps. It is not a replacement for a figure to summarize all types of resistance.
-The table is too long and not very helpful in its current format. It could be an appendix and summarized one should be included in the manuscript
-Finally I did not check for use of AI or other types of plagiarism. But this is an excellent review overall.
Author Response
Dear Reviewer
First of all, we appreciate your positive and constructive comments.
This is an excellent review on resistance mechanisms of Acinetobacter.
The manuscript is comprehensive and informative.
The organization is poor. The items are so long that the reader could get lost. -I suggest more titles under 3. virulence
Thank you very much for your effective comments. Your comments have been considered in our manuscript. In this regard, we added subtitles to this section (Pages 4-9).
-Also suggest a simple figure to show all the mechanisms of resistance with their commonality and subtypes.
Thank you very much for your effective comments. Your comments have been considered in our manuscript (Page 10, figure 1).
-Figure 1 is excellent but not comprehensive and does not help more than showing the location of the efflux pumps. It is not a replacement for a figure to summarize all types of resistance.
Thank you very much for your effective comments. Your comments have been considered in our manuscript (Page 10, figure 1).
-The table is too long and not very helpful in its current format. It could be an appendix and summarized one should be included in the manuscript
Thank you very much for your effective comments. Your comments have been considered in our manuscript (Page 5, table 1).
-Finally I did not check for use of AI or other types of plagiarism. But this is an excellent review overall.
Thank you very much. No, AI recruitment or copy-paste have been done.